MSUHEP-21-023, PITT-PACC-2117, SMU-HEP-21-09
FERMILAB-CONF-21-361-QIS-SCD-T

# NNLO constraints on proton PDFs from the SeaQuest and STAR experiments and other developments in the CTEQ-TEA global analysis

Marco Guzzi[1], T. J. Hobbs[2,3], Tie-Jiun Hou[4], Xiaoxian Jing[5], Keping Xie[6],
Aurore Courtoy[7], Sayipjamal Dulat[8], Jun Gao[9], Joey Huston[10], Pavel M. Nadolsky[5*],
Carl Schmidt[10], Ibrahim Sitiwaldi[8], Mengshi Yan[11], and C.-P. Yuan[10]

**1** Kennesaw State University, Kennesaw, GA 30144, U.S.A.
**2** Fermi National Accelerator Laboratory, Batavia, IL 60510, U.S.A.
**3** Illinois Institute of Technology, Chicago, IL 60616, U.S.A.
**4** Northeastern University, Shenyang 110819, China
**5** Southern Methodist University, Dallas, TX 75275-0181, U.S.A.
**6** University of Pittsburgh, Pittsburgh, PA 15260, U.S.A.
**7** Instituto de Física, Universidad Nacional Autónoma de México, CDMX 01000, Mexico
**8** Xinjiang University, Urumqi, Xinjiang 830046 China
**9** Shanghai Jiao Tong University, Shanghai 200240, China
**10** Michigan State University, East Lansing, MI 48824 U.S.A.
**11** Peking University, Beijing, 100871, China
*Email: nadolsky@smu.edu

August 14, 2021

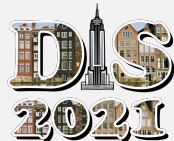

*Proceedings for the XXVIII International Workshop
on Deep-Inelastic Scattering and Related Subjects,
Stony Brook University, New York, USA, 12-16 April 2021*
doi:10.21468/SciPostPhysProc.xx.xxxx

## Abstract

We review progress in the global QCD analysis by the CTEQ-TEA group since the publication of CT18 parton distribution functions (PDFs) in the proton. Specifically, we discuss comparisons of CT18 NNLO predictions with the LHC 13 TeV measurements as well as with the FNAL SeaQuest and BNL STAR data on lepton pair production. The specialized CT18X PDFs approximating saturation effects are compared with the CT18sx PDFs obtained using NLL/NLO small-$x$ resummation. Short summaries are presented for the special CT18 parton distributions with fitted charm and with lattice QCD inputs. A recent comparative analysis of the impact of deuteron nuclear effects on the parton distributions by the CTEQ-JLab and CTEQ-TEA groups is summarized.

## 1  Introduction

Unpolarized parton distribution functions (PDFs) provide detailed models of the hadron structure for QCD computations at collision energies from a few GeV in fixed-target experiments to thousands of GeV at the Large Hadron Collider (LHC). The global QCD analysis is a systematic method to determine the PDFs and their uncertainties from precise cross

section measurements in deep-inelastic scattering (DIS) as well as production of vector bosons, jets, and massive quarks. Modern global analyses of PDFs test self-consistency of incisive perturbative QCD computations with dozens of scattering experiments. The CTEQ-Tung Et Al. (CTEQ-TEA) group has released its recent general-purpose NNLO PDF ensemble CT18 [1] based on the analysis that included eleven new data sets from LHC at $\sqrt{s} = 7$ and 8 TeV, available by mid-2018, as well as non-LHC data. Since their publication, the CT18 PDFs were confronted with the latest measurements at LHC, RHIC, and FNAL SeaQuest experiment. At the DIS'2021 workshop, the CTEQ-TEA group presented several studies of the latest results using the versatile CT18 framework. This proceedings contribution will summarize some of these studies after reviewing the key features of the CT18 analysis.

In addition to this contribution, our group presented other contributions to the DIS'2021 proceedings. A new CT18QED NNLO analysis with two implementations of the photon PDF was released in June 2021 [2,3]. Impact of the combined heavy-quark HERA DIS data on CT18 PDFs is discussed in [4]. New CT18 NNLO PDFs with fitted charm (CT18 IC) are introduced in Sec. 5. The large-$x$ falloff behavior of proton PDFs has been studied in the context of the CT18 NNLO fits [5,6]. A CT18 fit that includes constraints from lattice QCD was presented in [7], as well as in Sec. 7. Finally, some implications of the CTEQ-TEA analyses for future DIS experiments are discussed in Ref. [8].

## 2 Key features of CT18 parton distributions

The CT18 global analysis [1] provides a default set of PDFs, dubbed "CT18 NNLO", which is recommended for a vast majority of applications. Extensive effort went into selecting the accurate new data sensitive to the PDFs as well as to test stability of the PDFs and their uncertainties under variations in our underlying assumptions and fitting methodology. Eleven new LHC data sets were selected for fitting by applying fast survey techniques, `ePump` [9] and `PDFSense` [10]. [The full fit of all such new data sets would be prohibitively computer-intensive.] In-house NLO fast interfaces, ApplGrid and fastNLO, were produced for the new experiments, supplemented by lookup tables for point-by-point NNLO/NLO K-factors. With these, a large number of NNLO fits were performed for various data selection choices and theoretical assumptions. The nominal PDF uncertainty of the CT18 ensemble accounts for a combination of experimental, theoretical, parametrization, and methodological uncertainties. A substantial part of this uncertainty reflects the choice of the PDF parametrization forms, examined using more than 250 trial functional forms in the run-up to the publication of the final PDF ensemble. The nominal uncertainty also covers the best-fit solutions obtained with alternative scale choices in some experiments.

In addition to requiring the total $\chi^2/N_{pt}$ to be close to unity, the CT18 PDFs were subjected to a number of *strong goodness-of-fit tests* [11]. The CT18 uncertainties, while moderately larger than those estimated by other groups, are robust in the sense that they largely cover the spread of central predictions obtained with different assumptions and selections of experiments. For example, mutual agreement of the data sets was examined using complementary approaches utilizing the effective Gaussian variables, Lagrange Multiplier scans, Hessian PDF updating, and $L_2$ sensitivity techniques. Based on these, only reasonably consistent data sets were included into the default CT18 fit. On the other hand, the ATLAS 7 TeV $W, Z$ production data set [12] was found to be in substantial tension with the NuTeV dimuon and HERA DIS data and thus was included only into the alternative CT18A and CT18Z fits.

Some choices, such as the inclusion of the ATLAS 7 TeV $W, Z$ data, an alterative

treatment of experimental correlated errors, or using an $x$-dependent scale in DIS cross sections, cf. Sec. 4, have resulted in the central PDFs that were sometimes significantly different from the nominal CT18 ones. These differences reflect the genuine uncertainties that remain even at the NNLO. Thus, while the nominal CT18 uncertainties cover the spread of the best-fit solutions under the majority of assumptions, we also provide an alternative fit, CT18Z, that is outside the nominal CT18 uncertainty in some cases. The CT18Z fit combines the choices made in two middle-of-the-road fits, CT18A and X, and results in the most extreme PDF deviations from CT18 tolerated by the data. For example, the CT18Z predictions for $gg \rightarrow$ Higgs production ($Z$ boson production) are lower by about 1% (higher by 3.7%) than the CT18 ones. With our moderately conservative prescription for the uncertainties, the CT18 and CT18Z uncertainties overlap at 90% probability.

## 3 CT18 predictions vs. new hadronic data

### 3.1 Impact of SeaQuest and STAR Drell-Yan pair production

**SeaQuest**. The recent data [13] released by the SeaQuest (E906) Experiment at Fermilab have stimulated considerable interest. Much of this owes to the potential sensitivity of the $\sigma^{pd}/\sigma^{pp}$ Drell-Yan cross-section ratio to the deviations from SU(2) flavor symmetry in the large-$x$ behavior of the nucleon PDFs. This can be seen by considering a leading-order calculation of the cross-section ratio in which one may derive the well-known approximation,

$$\frac{\sigma^{pd}}{2\sigma^{pp}} \approx \frac{1}{2}\left(1 + \frac{\bar{d}(x_t)}{\bar{u}(x_t)}\right) \ , \tag{1}$$

where $x_t$ is the PDF momentum fraction of the fixed target. Flavor-symmetry breaking is a signature of nonperturbative QCD dynamics — in this case, signalled by $\bar{d} \neq \bar{u}$, particularly in the region $x > 0.1$. Various theoretical models (see, *e.g.*, Ref. [14]) generally predict a high-$x$ excess of $\bar{d}$ relative to $\bar{u}$ such that the ratio $\bar{d}/\bar{u} \geq 1$ in this region. For instance, the violation of flavor SU(2) may be realized through pion emission and reasorbtion (*i.e.*, pion cloud) contributions to the proton wave function. This scenario leads naturally to $\bar{d}/\bar{u} \geq 1$ given the preferred dissociation of the proton: $p \rightarrow \pi^+[u\bar{d}] + n[ddu]$. In this context, the behavior of the somewhat older high-$x_t$ E866 ratios [15] (and the two highest $x_t$-bins, in particular) pose a challenge to many theoretical models, as they have typically been found to favor a downturn in the extracted $\bar{d}/\bar{u}$ PDF ratio once fitted in full QCD global analyses.

The newer SeaQuest data extend the cross-section ratio data measured earlier by E866 to somewhat higher $x_t \sim 0.45$ with enhanced precision. At the same time, SeaQuest also reports measurements in a kinematical region intersecting the coverage of E866 over the approximate range $0.15 \lesssim x_t \lesssim 0.35$. In this meeting, we report a first study of the impact of the SeaQuest data within the CT global analysis. We find the SeaQuest data to be in strong agreement with theory predictions based on CT18 NNLO *before fitting*, with $\chi^2_E/N_{pt} = 0.82$. This partly reflects the parametrization choices made in CT18 [1] for the high-$x$ behavior of $\bar{d}, \bar{u}$, which are selected to preserve $\bar{d}/\bar{u} \geq 1$ at high $x$ on the QCD modeling logic discussed above. The SeaQuest ratios thus prefer the $\bar{d}/\bar{u} \geq 1$ high-$x$ behavior favored by nonperturbative QCD-motivated models discussed above.

In Fig. 1 (left) we present the fitted $\bar{d}/\bar{u}$ PDF ratio at $Q = 2$ GeV in three NNLO analyses: the baseline CT18 NNLO result ("CT18", in blue); a fit in which the new SeaQuest data are included with the default CT18 data set ("CT18sq", in red); and a new alternative fit ("CT18n", in green), in which we replace the E866 ratio data with

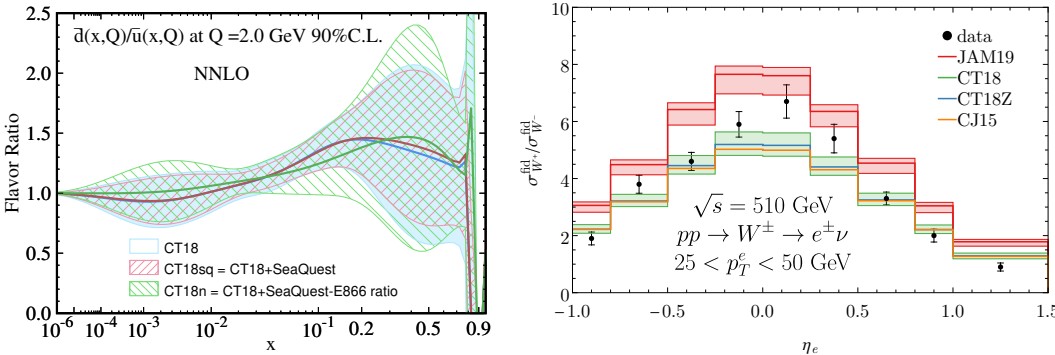

Figure 1: (Left) The NNLO $\bar{d}/\bar{u}$ PDF ratios at the scale $Q = 2$ GeV from two preliminary fits of the SeaQuest ratio data, in addition to the baseline CT18 NNLO result. (Right) A preliminary calculation of the cross-section ratio of $W^+$ to $W^-$ at RHIC $\sqrt{s} = 510$ GeV, using the recent CT18(Z) [1], CJ15 [16], and JAM19 [17] PDFs, compared with the STAR measurement [18].

SeaQuest, while simultaneously applying a fixed nuclear correction to the deuteron DIS data [19], removing the inclusive $\nu A$ DIS data, and, for the first time, including an overall 5% uncorrelated uncertainty to account for nuclear effects in the E605 Drell-Yan data on copper. In line with the robust agreement of the CT18 NNLO predictions with SeaQuest, fitting the SeaQuest ratios in CT18sq leads to a modest enhancement in the $\bar{d}/\bar{u}$ ratio at $x > 0.1$ and a mild corresponding reduction in the PDF uncertainty. This behavior is reinforced at very high $x \geq 0.3$ by the data-set modifications in CT18n with a slightly larger suppression of the PDF ratio for $0.05 \leq x \leq 0.3$.

It is also instructive to investigate the pulls of select data sets in the CT18n global fit on $\bar{d}/\bar{u}$ using the $L_2$ sensitivity method [20], as shown in Fig. 2. For any PDF-dependent QCD observable of interest, the $L_2$ sensitivity provides an estimate of the $\Delta \chi^2$ for a given experiment when the PDFs are increased by $+1\sigma$ in the direction associated with the PDF uncertainty for this observable. The $L_2$ sensitivity technique uses the published error PDFs and is very fast, in contrast to the slow computations of $\Delta \chi^2$ during the PDF fit itself. By computing the $L_2$ sensitivities to the PDFs, $f_a(x, Q)$, at a given $x$ and $Q$ for each fitted data set, we obtained a common metric to quantify the strength of statistical pulls on the PDFs in various fits involving the SeaQuest data.

In Fig. 2 (left), we show the $L_2$ sensitivity of the data fitted in CT18sq to $\bar{d}/\bar{u}$ at $Q = 2$ GeV. In this case, fitting the SeaQuest (Expt. 206) and E866 (Expt. 203) DY ratios in conjunction produces significant high-$x$ tension between the two, especially for $x \gtrsim 0.2$. This behavior is unsurprising, given the apparent tensions at highest $x_t$ in the $\sigma^{pd}/\sigma^{pp}$ cross sections measured by SeaQuest and E866.

For this reason, we examine the pattern of tensions in the alternative CT18n NNLO fit in which we replace the E866 ratios with those of SeaQuest. In this scenario, the robust consistency of the SeaQuest data with theory predictions in the absence of competing pulls from the E866 ratio data is such that SeaQuest has very small tension ($|\Delta \chi_E^2| \lesssim 1$) with other fitted data sets. As a result, they are not plotted. At the same time, intriguing evidence of some competing pulls at high $x$ emerge, especially between the absolute $\sigma^{pp}$ E866 cross sections (Expt. 204) and the deuteron-corrected NMC ratio data (Expt. 118). These early findings suggest the importance of further investigation of the interplay of nuclear data sets in fits and the potential role that the SeaQuest data may play in resolving data-set pulls on $\bar{d}/\bar{u}$.

**STAR**. Parallel to these developments, the STAR experiment at Brookhaven's RHIC

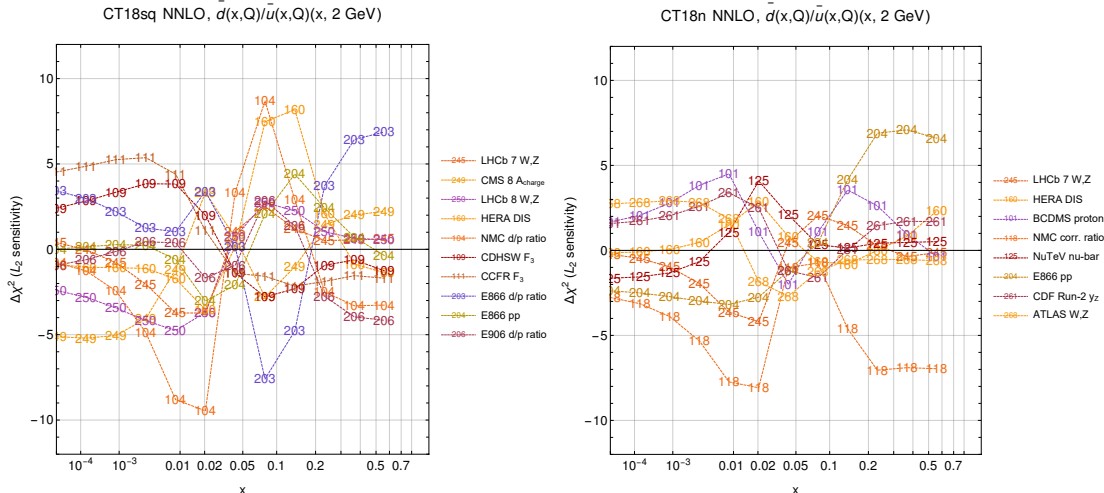

Figure 2: (Left) The preliminary $L_2$ sensitivity of select experiments to $\bar{d}/\bar{u}$ from the CT18sq NNLO fit in which the SeaQuest data are added to the rest of the CT18 global data, including the E866 $d/p$ ratio. (Right) An analogous $L_2$ sensitivity calculation, but for an alternative fit, CT18n NNLO, involving a number of alterations in the fitted nuclear data; here, the SeaQuest ratio replace the E866 ratio as explained in-text. In this scenario, the SeaQuest data exhibit no significant tensions with other experiments sensitive to $\bar{d}/\bar{u}$ and therefore do not appear in the right panel. On the other hand, at high $x$ the E866 $\sigma^{pp}$ absolute cross-section data (Expt. 204) now pull against the NMC ratio data with the deuteron correction (Expt. 118).

facility recently announced [18] new measurements of inclusive electroweak (EW) boson production in $pp$ Drell-Yan at $\sqrt{s} = 510$ GeV. It is appropriate to consider these data along with the SeaQuest measurements discussed above, given the potentially complementary sensitivity of the STAR data to the $\bar{u}$ and $\bar{d}$ PDFs due to the charge structure of the EW boson's interactions with the proton's flavor currents. As a first study, we therefore briefly report theory predictions for the $W^+/W^-$ cross section ratio released by STAR.

In Fig. 1 (right), we compare the recent STAR data to predictions based on several PDF sets, including CT18 and CT18Z NNLO, shown in green and blue, respectively. We generally obtain qualitatively reasonable agreement with the reported pseudorapidity distribution, although with a moderate underprediction of the measurement at very central values of $0 \lesssim \eta_e \lesssim 0.5$. The *shape* of the $\eta_e$ is relatively well reproduced by our theory predictions, however, and we obtain for CT18 a value of $\chi^2_E/N_{\rm pt} = 2.9$, assuming the systematic uncertainties are uncorrelated. This is the case due to then fact that correlated sources of systematic uncertainty cancel in the cross-section ratio. A significant share of the total $\chi^2_E$ in this case is attributable to the description of a small number of bins at very central and highest $\eta_e$. As the PDFs in the relevant $x$ region are already constrained well by other experiments, inclusion of the STAR data in the fit will require further tests of mutual consistency. The possible PDF sensitivity of the STAR Drell-Yan measurements is such that they merit further consideration in global fits involving the SeaQuest ratios and other high-quality hadronic data.

## 3.2 Comparisons of CT18 NNLO PDFs with latest LHC data

Measurements of inclusive cross sections at hadron colliders serve as benchmark tests of the Standard Model. Our CT18 publication [1] included many predictions for the standard-candle cross sections based on the CT18(A/X/Z) PDFs. Compared to the absolute cross

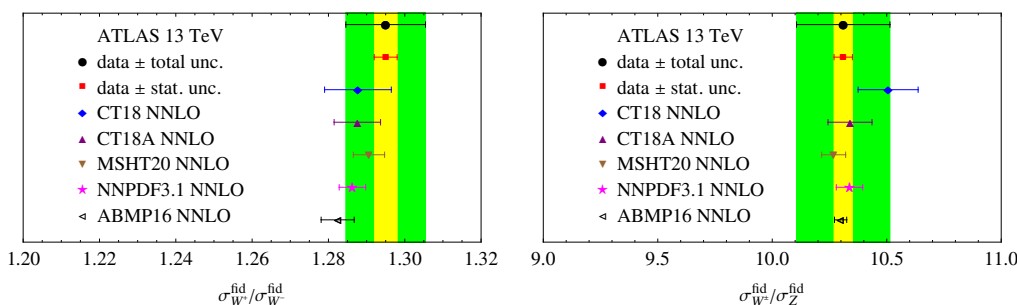

Figure 3: Predictions for fiducial cross section ratios of $W^+$ to $W^-$ boson (left) and $W^{\pm}$ to $Z$ boson (right), compared to the ATLAS measurements [21] measurement at 13 TeV.

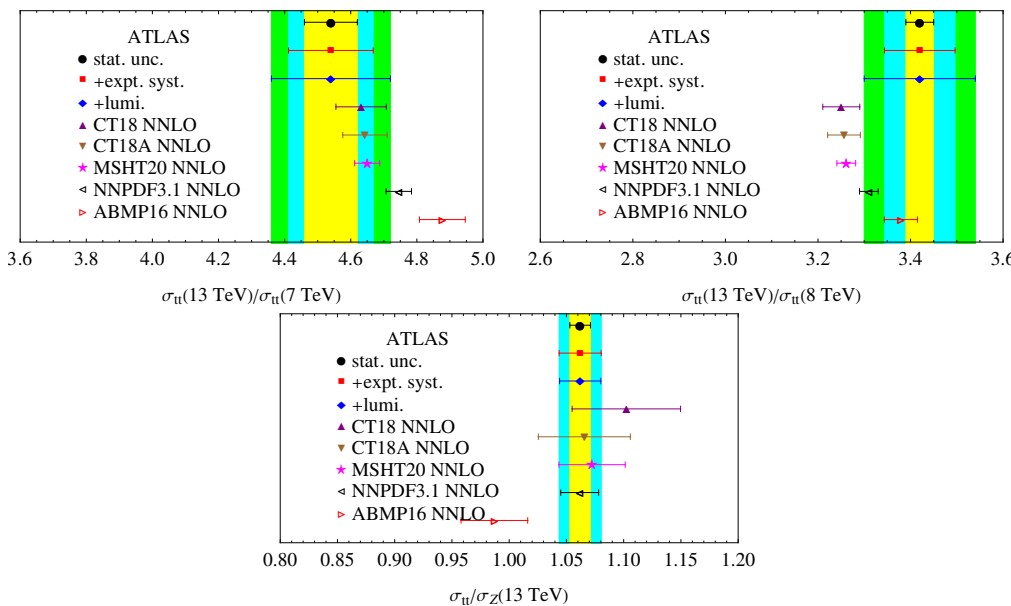

Figure 4: Top-antitop production cross section ratios for $\sqrt{s} = 13, 7, 8$ TeV and the ratio $\sigma_{t\bar{t}}/\sigma_Z$ at 13 TeV, compared with the ATLAS measurements [22].

sections, the cross section ratios may offer an advantage of a higher precision because of the cancellation of correlated uncertainties. In this proceedings, we will focus on the ratios of fiducial cross sections of $W^{\pm}, Z$ and $t\bar{t}$, and compare them with the ATLAS measurements at 13 TeV [21, 22]. The PDF uncertainties are given at the 68% confidence level (CL) in the next two figures.

First, Fig. 3 shows the fiducial cross section ratios for $W^{\pm}$ and $Z$ production in comparison to the ATLAS data [21]. The theoretical predictions are calculated with the NLO `APPLgrid` [23] and combined with the NNLO/NLO $K$-factors obtained with `MCFM` [24]. We see that for the $W^+/W^-$ case, the predictions from all the included PDFs [1, 25–27] are on the lower side compared with the data. For the $W^{\pm}/Z$ ratio, the predictions are in a good agreement with the data, except that CT18 is on the margin of the experimental uncertainty, while CT18A agrees well. Compared with the other PDFs, the CT18(A) give somewhat larger uncertainties because of the choices discussed in Sec. 2. The CT18A uncertainty is slightly smaller than the CT18 one because of the smaller uncertainty on the strangeness PDF at $x \approx 0.02$ resulting from competing pulls from the ATLAS 7 TeV $W^{\pm}/Z$ data [12] and NuTeV dimuon data, as discussed in detail in Appendix C in Ref. [1].

Next, Fig. 4 presents cross section ratios for $t\bar{t}$ production, calculated at NNLO with `Top++` [28]. The upper row shows the ratios of $t\bar{t}$ total cross sections at center-of-mass

energies 7, 8, and 13 TeV. The lower row shows the ratio of $t\bar{t}$ and $Z$ total cross sections at $\sqrt{s} = 13$ TeV. Interestingly, the predictions of $t\bar{t}$ cross section ratios of 13-to-7 TeV and 13-to-8 TeV pull in different directions with respect to the ATLAS measurements [22]. That is, the theoretical predictions for the ratios $\sigma_{t\bar{t}}(13 \text{ TeV})/\sigma_{t\bar{t}}(7 \text{ TeV})$ and $\sigma_{t\bar{t}}(13 \text{ TeV})/\sigma_{t\bar{t}}(8 \text{ TeV})$ tend to lie above and below the data, respectively. Some predictions are even out of the experimental error bands, even considering the PDF uncertainty, *e.g.*, for ABM16 PDFs [27]. Compared to the ratios at different energies, the measurement of ratio $t\bar{t}$-to-$Z$ at 13 TeV is much more accurate, largely due to the cancellation of luminosity uncertainty. We see that the central value of CT18 prediction is on the higher side, while the ABMP16 is on the lower side. With the PDF uncertainty, the CT18 prediction can cover the experimental data, while ABMP16 cannot. Same as before, the CT18(A) gives larger error bands due to its moderately conservative tolerance criterion, as discussed in Sec. 2.

## 4    CT18X saturation and CT18sx small-x resummation

In DIS at Bjorken $x_B < 10^{-3}$ and momentum transfer $Q$ of a few GeV, on the general grounds one expects enhanced small-$x$ logarithmic contributions and eventually partonic saturation. In the fixed-order NNLO fits, the factorization scale dependence of the DIS cross sections becomes large as $x_B$ becomes small. By choosing a factorization scale in DIS that depends on $x_B$ in a fixed-order fit [1], one can obtain the PDFs that are similar to those from the fits that implement small-$x$ corrections.

To complement the default general-purpose CT18 PDF ensemble, our group has published two alternative ensembles, CT18X and CT18Z, in which the $x$-dependent factorization scale for the DIS cross sections was set to $\mu_{F,\text{DIS}}^2 = 0.8^2 \left(Q^2 + (0.3 \text{ GeV}^2)/x_B^{0.3}\right)$. The form of $\mu_{F,\text{DIS}}^2$ is motivated by the partonic saturation [29], and its parameters were determined from a fit. In the $x$-$Q$ region accessed at HERA, the modifications in the CT18X PDFs with respect to CT18 are like the ones observed in the fits with the next-to-leading-logarithmic (NLLx) BFKL resummation [30, 31]. Yet, in cosmic-ray experiments and future DIS at $x_B$ below $10^{-5}$, one generally expects the nonlinear saturation to behave differently from its BFKL approximation [32]. It is therefore important to have several PDF models to estimate the range of theoretical predictions at the smallest $x_B$.

To explore this issue, we used the public package HELL [33, 34] interfaced with APFEL [35] to obtain CT18sx PDFs by evolving the initial CT18 NNLO PDFs from the initial scale 1.3 GeV using the NLLx+NNLO, rather than NNLO evolution. We then computed the small-$x$ resummed (NLLx) structure functions $F$ ($\equiv F_2$ or $F_L$) for the CT18sx fit by starting from the NNLO ones in the SACOT-$\chi$ [36, 37] heavy-quark scheme with a $K$-

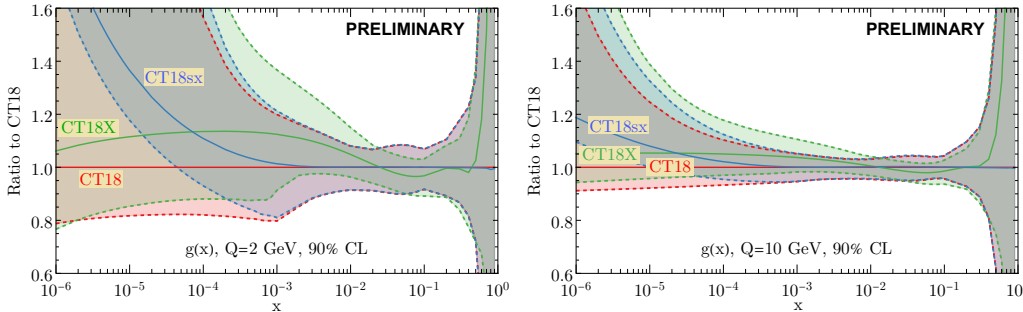

Figure 5: CT18, CT18X, and CT18sx NNLO gluon PDFs at $Q = 2$ and 10 GeV.

factor approach:

$$F^{\text{NLLx, SACOT}}(\text{CT18sx}) = F^{\text{NNLO, SACOT}}(\text{CT18}) \underbrace{\frac{F^{\text{NLLx}}(\text{CT18sx})}{F^{\text{NNLO}}(\text{CT18})}}_{\equiv K_{1},\text{FONLL}} . \qquad (2)$$

The factor $K_1$, accounting for the difference between the NLLx(+NNLO) and NNLO predictions in the FONLL-C heavy-quark scheme [38], was computed in APFEL. At small $x$ relevant in this study, this factor depends little on either the PDF or the heavy-quark scheme. With this method, we find that the CT18sx small-$x$ resummed structure functions closely agree with the fixed-order CT18 and "saturation" CT18X structure functions in the fitted region, yet have different extrapolations at $x \to 0$.

Figure 5 compares CT18, CT18X, and preliminary CT18sx gluon PDFs at $Q = 2$ and 10 GeV. We see that the small-$x$ resummation of CT18sx enhances the gluon PDF at $x \lesssim 10^{-3}$, while the large-$x$ region remains unchanged. A similar enhancement above CT18 occurs at moderate $x$ in CT18X at $Q = 2$ GeV, but the growth is tamed toward $x \to 10^{-6}$, in qualitative agreement with the saturation expectations. The CT18sx and CT18X gluons become similar as $Q$ increases, as illustrated for $Q = 10$ GeV in the right Fig. 5.

Many implications about the small-$x$ dynamics can be gleaned from comparing the CT18sx and CT18X predictions. The differences among the structure functions are less striking than in the PDFs, as the PDF differences are partly compensated by the changes in the corresponding Wilson coefficients. The differences are more significant in the longitudinal structure function $F_L$ than in the transverse $F_2$, as we illustrate in Fig. 6 at a low scale $Q^2 = 3.5$ GeV$^2$. In $F_2(x_B)$, at such $Q^2$, both CT18X and preliminary CT18sx predict a similar amount of enhancement above CT18 at a low $x_B$. However, in $F_L(x_B)$ for CT18X, the initial enhancement above CT18 at $x_B \sim 10^{-2}$ is eventually flattened or even suppressed at $x_B \lesssim 10^{-4}$, with the specific $x_B$ dependence being tunable according to the $x_B$-dependent scale. In contrast, the CT18sx fit always predicts a larger $F_L$ than at the fixed-order NNLO, and the difference only grows with $x$ deceasing. It would be thus very interesting to have a future experiment that can distinguish between the growing or flattening trends for $F_L(x_B)$ at low $x_B$ values.

As the final remarks in this section, Fig. 6 shows that both the SACOT-$\chi$ and FONLL-C schemes lead to comparable trends for $F_{2,L}$ at $x_B \lesssim 10^{-4}$. The small-$x$ enhancement of the structure functions, or reduction in the CT18X $F_L(x_B \lesssim 10^{-4})$ case, only occur in the low $Q^2$ region and die out as $Q^2$ increases. Higher-twist contributions at small $x_B$ [39], not included here, can produce comparable effects.

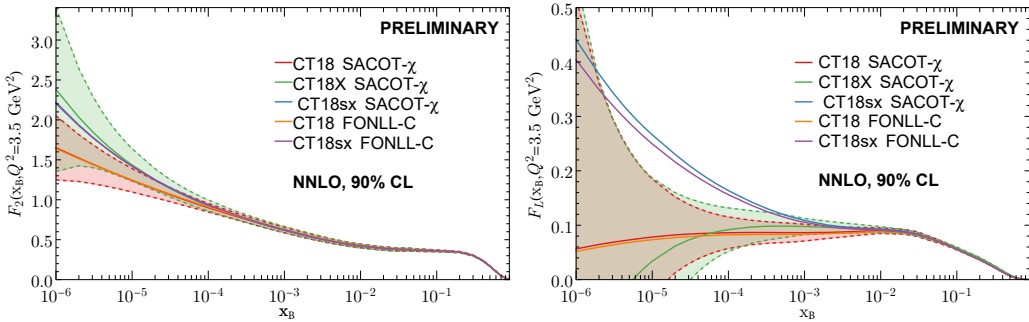

Figure 6: The comparison of structure functions $F_{2,L}(x, Q^2)$ at $Q^2 = 3.5$ GeV$^2$ among CT18, CT18X, and CT18sx.

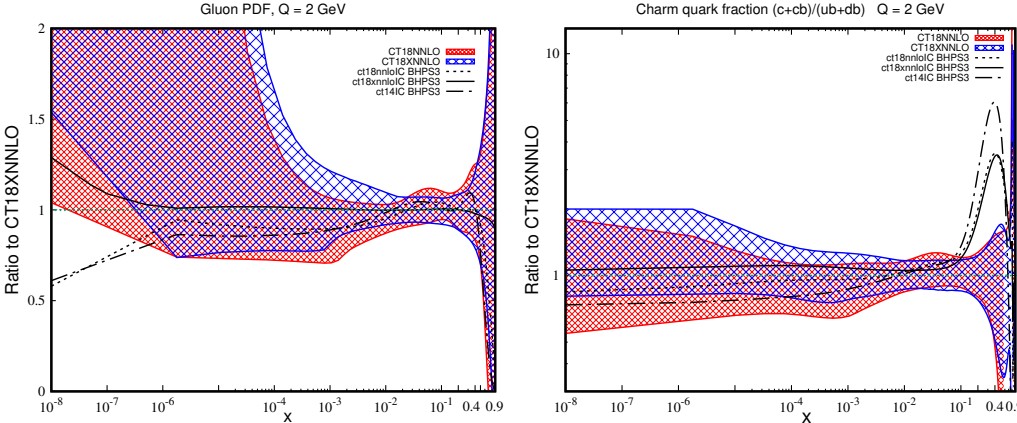

Figure 7: (Left) Gluon PDF. (Right) Charm-quark fraction ratio. The error bands represent the CT18NNLO (red) and CT18XNNLO (blue) PDF uncertainties at 90% C.L. [1]. The solid line is CT18XNNLO-IC/CT18XNNLO, the dashed line is CT18NNLO-IC/CT18XNNLO, and dot-dashed line is CT14NNLO-IC/CT18XNNLO.

## 5   Hadronic scattering processes with nonperturbative charm

As new global PDFs analyses are challenged by new measurements at the LHC that grow increasingly precise, it is important to study the effect of PDF correlated parameters (e.g., heavy-quark masses, kinematic suppression near threshold, hard scales in the cross section calculation, and power-suppressed contributions), whose impact competes in magnitude with higher-order QCD perturbative corrections. As an additional study performed within the CT18 framework, we explored the impact of nonperturbative contributions from power-suppressed scattering processes in DIS. In particular, we studied the impact on the gluon and charm fraction ratio at large $x$ from intrinsic charm (IC) production. IC is introduced as a phenomenological parametrization ("fitted charm") that is determined in a global QCD analysis as an independent PDF functional form [40–43]. We recall [42] that the fitted charm PDF has a process-dependent component that can be partly traced to power-suppressed radiative contributions in DIS that do not necessarily coincide at the LHC. The dynamical origin of this nonperturbative contribution and its magnitude have been associated with the excited $|uudc\bar{c}\rangle$ Fock states of the proton wave function [44–50]. The range of validity of nonperturbative charm models has been studied in the previous CT14 IC [42] analysis, where an estimate of the IC magnitude is done under the assumption that this contribution is factorized in DIS processes.

Here, we report preliminary results of a global fit obtained with the CT18 data ensemble and including the BHPS3 parametrization [42] that realizes the model from [50]. For the BHPS3 model, the charm probability integral is defined as in the original BHPS model [44, 45], but solved numerically and keeping the exact dependence on the proton and quark masses. In Fig. 7, we show ratio plots illustrating the impact of BHPS3 IC on the CT18 and CT18X gluon PDF and charm-fraction ratio evaluated at $Q = 2$ GeV. The result of the CT14IC fit is also shown. The error bands are evaluated at the 90% confidence level. In the right subfigure, the charm-quark fraction at $0.1 \leq x \leq 0.5$ is enhanced less in the CT18IC and CT18XIC scenarios as compared to CT14IC. The difference between these global fits has implications for prompt charm production in proton-proton collisions, which has the potential to discriminate among nonperturbative charm models. A recent study [51] analyzes the experimental data on charm meson production measured

by LHCb at 7 and 13 TeV [52,53] by using NLO theory predictions obtained in the recently developed SACOT general-mass factorization scheme with massive phase space (SACOT-MPS) [54]. SACOT-MPS is an amended version of the S-ACOT scheme [36, 37, 55–58] applied to the case of proton-proton collisions. Forthcoming analyses will assess the impact of IC by using SACOT-MPS and the new prompt charm production measurements at LHCb.

# 6   Deuterium scattering experiments in CJ and CT analyses

The CT18 global analysis [1] reached an important, not initially anticipated, observation that the constraints on the PDFs from fixed-target data remain very influential even after the inclusion of LHC data sets. In particular, DIS on deuterium targets by BCDMS and NMC plays the key role in separating up- and down-quark PDFs. As a result, the measurement of the weak mixing angle at the LHC and other measurements sensitive to flavor separation depend on the treatment of nuclear effects in the PDF fits of the deuteron DIS data. The roles of nuclear and target-mass effects in CTEQ-JLab (CJ) and CT NLO fits were compared side-by-side in [19] using the $L_2$ sensitivity method [20]. As summarized in Sec. 3.1, the $L_2$ method allows us to compare, on an apples-to-apples basis, the PDF pulls of the data sets fitted in the distinct CJ and CT fitting frameworks. By comparing various implementations of the deuteron correction, we showed that the freely-fitted deuteron corrections modify the PDF uncertainty at large momentum fractions. In addition, due to the influence of sum rules and nontrivial correlations among the PDFs of different flavors, deuteron corrections to DIS structure functions at large $x$ have important secondary effects on, e.g., the gluon or sea-quark PDFs over a range of $x$, as well as the $d_{val}$ distribution at lower $x \sim 0.03$, of relevance to precision studies in the electroweak sector. Hence, in order to achieve ultimate precision in tests of the SM in the electroweak sector, it is critical to have a correct treatment of the deuteron corrections in future PDF analyses.

# 7   PDFs with lattice QCD inputs

In addition to the projects reported above, we have also explored a global analysis by taking a prediction from a lattice QCD calculation, which is dubbed as the CT18CS fit. Its preliminary version has been reported at this workshop as the CT18CSpre fit [7]. This study was motivated by the Euclidean path-integral formulation of the hadronic tensor of the nucleon, which uncovered that there are two kinds of sea partons, associated with connected and disconnected lattice configurations. By assuming distinct small-$x$ behaviors for these two sea parton components, and imposing a constraint from the lattice calculation on the ratio of the strange momentum fraction to that of the $\bar{u}$ or $\bar{d}$ in the disconnected insertion, we obtained the CT18CSpre fit. We found that the qualities of the CT18CSpre fit (with some specific assumption and ansatz imposed on various sea components) and the standard CT18 NNLO are comparable, though CT18CSpre has more parton degrees of freedom as compared to CT18. This new fit, CT18CSpre, allows lattice calculations of separate flavors in both the connected and disconnected insertions to be directly compared with the global analysis results term-by-term.

## 8 Conclusions

In this contribution, we reviewed several follow-up studies pursued by the CTEQ-TEA group since the publication of the CT18 global QCD analysis. As usual, tables of CT PDFs are available for downloading from the LHAPDF library. The CTEQ-TEA website on HEPForge provides tables of PDFs and supplemental figures. Preliminary PDF fits discussed in this contribution may be available by request from authors.

## 9 Acknowledgements

We are grateful to Markus Diefenthaler and Paul Reimer of SeaQuest and Matt Posik of STAR for helpful exchanges related to the interpretation of these recent Drell-Yan data sets.

## 10 Funding information

The work on this contribution is supported in part by the US Department of Energy under Grants No. DE-FG02-95ER40896, DE-SC0010129; by the National Science Foundation under Grants No. PHY-1820818, PHY-1820760, PHY-2013791; and by PITT PACC. A.C. is supported by UNAM Grant No. DGAPA-PAPIIT IA101720 and CONACyT Ciencia de Frontera 2019 No. 51244 (FORDECYT-PRONACES). T.J. Hobbs acknowledges support from the Fermi National Accelerator Laboratory, managed and operated by Fermi Research Alliance, LLC under Contract No. DE-AC02-07CH11359 with the U.S. Department of Energy as well as from a JLab EIC Center Fellowship. J.G. was supported by the National Natural Science Foundation of China under Grants No. 11875189 and No. 11835005. C.P.Y. is also grateful for the support from the Wu-Ki Tung endowed chair in particle physics.

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
