# Peer review of "NNLO constraints on proton PDFs from the SeaQuest and STAR experiments and other developments in the CTEQ-TEA global analysis"

_SciPost Physics Proceedings_

## Round 1 · Referee Report · Anonymous (Referee 1) · 2021-9-20

Strengths

This is a concise summary of a series of recent studies performed by the CTEQ-TEA collaboration on the determination of the parton distributions functions of the proton. The spectrum of topics is particularly wide and focus on developments initiated and/or completed after the release of their latest global analysis, CT18.

Weaknesses

The summary is sometimes a little unbalanced, in that results already published in dedicated papers are discussed in detail, while new results are occasionally a little sketchy.

Report

This contribution to the DIS2021 conference proceedings presents a summary on a series of recent studies performed by the CTEQ-TEA collaboration about the determination of the parton distribution functions (PDFs) of the proton. After a brief recap on the key features of their latest global analysis, CT18, the authors present theoretical predictions for recent RHIC, FNAL and LHC measurements, then they discuss variants of the nominal CT18 analysis in which the effects of small-x resummation, nonperturbative charm, nuclear corrections in deuterium targets and lattice QCD data are assessed.

The contribution addresses several topics, some of which have been the object of specific studies, already presented in published papers, and some of which are presented here in the form of preliminary results for the first time. Sometimes I find the discussion a little unbalanced, in that the old material is discussed in more detail than the new one. Although I would have appreciated some of the new results to be presented more extensively (specifically those in Sects. 5 and 7), I understand that the nature of the contribution, the fact that some results are still preliminary and space constraints did not allow the authors to be more detailed. That being said, I would be grateful if the authors could address the minor points that I list in the next section. In my opinion, these will help clarify some aspects of the various analyses. Once this is done, the manuscript can be published in SciPost Physics Proceedings.

Requested changes

1- Page 2, line 8 from the top. The authors state: "Since their publication, the CT18 PDFs were confronted with the latest measurements at LHC, RHIC, and FNAL SeaQuest experiment." Can they provide a reference as to where this has been discussed? Or can they make it clear that RHIC and FNAL measurements are discussed in this contribution for the first time? 2- Page 2, line 5 from the beginning of Sect. 2. The authors state: "Eleven new LHC data sets were selected for fitting by applying fast survey techniques". Do I understand correctly that these eleven datasets were selected out of many more? Can the authors clarify? 3- Page 2, line 11 from the beginning of Sect. 2. The authors state: "The nominal PDF uncertainty of the CT18 ensemble accounts for a combination of experimental, theoretical, parametrization, and methodological uncertainties." Can they clarify the nature of the theoretical uncertainties represented in their PDF uncertainties? 4- Page 3, line 8 below Eq. (1). I presume that $x_t$, which is never defined in the text, is the target momentum fraction carried by the struck parton. Can the authors please clarify the meaning of $x_t$? 5- Sect. 3.1. If I understand correctly, the authors fit the $\bar d$/$\bar u$ ratio, which is reconstructed from the measurement of $\sigma^{pd}/2\sigma^{pp}$ by means of the approximation given in Eq. (1). Is my understanding correct? Why do they not use directly the cross section ratio? Can they please clarify this point? 6- Page 4, first line. The authors state that they perform a variant fit in which they "replace the E866 ratio data with SeaQuest, while simultaneously applying a fixed nuclear correction to the deuteron DIS data". This statement seems to suggest that the authors fit the SeaQuest cross section ratio, as they did with E866, but this contradicts my understanding (see above). Also, if they fit the cross section ratio, are they considering a nuclear correction due to the deuteron target also in the SeaQuest measurement or only in the DIS data, as stated? Is this not a little inconsistent? 7- Page 4, line 20. I read that "fitting the SeaQuest and E866 DY ratios in conjunction produces significant high-x tension between the two". This conclusion is made from the analysis of the L2 sensitivity plot reported in Fig.2. Is this statement in contradiction with what I read on Page 3: " We find the SeaQuest data to be in strong agreement with theory predictions based on CT18 NNLO". Do CT18 NNLO PDFs not include E866 data? Can the authors please clarify? 8- Figures 3-4. Can the authors please use the same symbols/colours for the same PDF sets displayed in the two figures? 9- Page 7, first line of Sect. 4. Can the authors clarify the difference between $x$, $x_t$ and $x_B$? The three notations are used across the manuscript. 10- I believe that I've spotted two typos. 1st line of page 4: simulataneously -> simultaneously; 26th line of page 8: deceased -> decreased.

  • validity: high
  • significance: high
  • originality: good
  • clarity: good
  • formatting: excellent
  • grammar: excellent

Author:  Pavel Nadolsky  on 2022-03-23  [id 2318]

(in reply to Report 1 on 2021-09-20)

Dear Editors,

we thank the referee for the attentive reading of the manuscript and helpful remarks. We are resubmitting the manuscript after addressing the referee's questions as listed below.

Best regards,

Pavel and the authors of the paper.

\documentstyle[12pt]{article} \begin{document} 1- Page 2, line 8 from the top. The authors state: "Since their publication, the CT18 PDFs were confronted with the latest measurements at LHC, RHIC, and FNAL SeaQuest experiment." Can they provide a reference as to where this has been discussed? Or can they make it clear that RHIC and FNAL measurements are discussed in this contribution for the first time?

--> The language here has been updated and clarified. Specifically, the modified text here now reads:

"Since their publication, the CT18 PDFs were confronted with the latest measurements from the LHC, RHIC, and FNAL SeaQuest experiments; in these proceedings, we present initial studies of data from SeaQuest and RHIC, which, until now, had not been investigated in the CT literature."

2- Page 2, line 5 from the beginning of Sect. 2. The authors state: "Eleven new LHC data sets were selected for fitting by applying fast survey techniques". Do I understand correctly that these eleven datasets were selected out of many more? Can the authors clarify?

--> In Sec. II.A.3 and elsewhere within the CT18 main paper, we discussed the selection of the newly-fitted LHC experiments in detail. Indeed, these were chosen from a larger collection of LHC experiments with potential PDF sensitivity; our ultimate selection was informed by the fast analysis methods we mention. To clarify this point, we have added an additional sentence here on page 2:

"The eleven newly-introduced LHC experiments were selected from a larger collection of more than three dozen LHC measurements which had been made available by 2018, and were targeted based on their PDF sensitivity as discussed in Sec.~II.A.3 of Ref.~\cite{Hou:2019efy}."

3- Page 2, line 11 from the beginning of Sect. 2. The authors state: "The nominal PDF uncertainty of the CT18 ensemble accounts for a combination of experimental, theoretical, parametrization, and methodological uncertainties." Can they clarify the nature of the theoretical uncertainties represented in their PDF uncertainties?

--> This matter is discussed in greater detail in the main CT18 paper; for instance, in Sec. II.A.5. In the course of producing CT18, various uncertainties were investigated, including those associated with scale choice, use of NNLO vs. resummation codes, and parametrization uncertainties. These effects were studied to ensure they are substantially covered by our PDF uncertainties. In addition, the effect of Monte Carlo integration uncertainty was directly included in the CT18 analysis. We have clarified these points by adding a new statement on Page 2:

"Among these, uncertainties due to the choice of perturbative scale, the selection of NNLO fixed-order vs.~resummation codes, the selected nonperturbative parametrization form, and Monte Carlo integration effects were carefully explored. These were either directly incorporated into the analysis, as with the Monte Carlo integration uncertainty, or examined to ensure that the computed PDF uncertainty encompassed variations associated with each of these choices."

4- Page 3, line 8 below Eq. (1). I presume that x t , which is never defined in the text, is the target momentum fraction carried by the struck parton. Can the authors please clarify the meaning of x t ?

--> Addressed following Eq. (3); language slightly modified.

5- Sect. 3.1. If I understand correctly, the authors fit the d ¯ / ū ratio, which is reconstructed from the measurement of σ pd /2σ pp by means of the approximation given in Eq. (1). Is my understanding correct? Why do they not use directly the cross section ratio? Can they please clarify this point?

--> We of course fit the released SeaQuest cross-section ratio data --- Eq. (1) is simply noted to indicate the expected sensitivity of the deuteron-to-proton ratio to dbar/ubar. To state this more clearly, we have revised the third sentence of the first full paragraph following Eq. (1):

"In this meeting, we report a first study of the impact of the SeaQuest data based on directly fitting the released cross-section ratios within the NNLO CT global analysis."

6- Page 4, first line. The authors state that they perform a variant fit in which they "replace the E866 ratio data with SeaQuest, while simultaneously applying a fixed nuclear correction to the deuteron DIS data". This statement seems to suggest that the authors fit the SeaQuest cross section ratio, as they did with E866, but this contradicts my understanding (see above). Also, if they fit the cross section ratio, are they considering a nuclear correction due to the deuteron target also in the SeaQuest measurement or only in the DIS data, as stated? Is this not a little inconsistent?

--> We clarified that we of course fit the SeaQuest ratio data directly as should always be done in a global analysis. Regarding the deuteron corrections, indeed these were only applied to the deuteron DIS data, not the Drell-Yan experiments. We would stress that these corrections represent precision effects which will be of greater importance in future analyses. We carry out an initial study of the impact of these deuteron corrections applied to DIS data along with the other nuclear modifications described in our manuscript. We intend to carry out further investigations of the compatibility of these treatments in future analyses. For now, we note this in some text we have added below Fig. 1:

"We introduce these modifications as a preliminary exploration of the influence nuclear effects can have on the statistical tensions among fitted data; while deuteron corrections were similarly analyzed for deuteron DIS data in Ref.~\cite{Accardi_2021}, we reserve a more comprehensive study of nuclear corrections within the CT framework to future work."

7- Page 4, line 20. I read that "fitting the SeaQuest and E866 DY ratios in conjunction produces significant high-x tension between the two". This conclusion is made from the analysis of the L2 sensitivity plot reported in Fig.2. Is this statement in contradiction with what I read on Page 3: " We find the SeaQuest data to be in strong agreement with theory predictions based on CT18 NNLO". Do CT18 NNLO PDFs not include E866 data? Can the authors please clarify?

--> The CT18 PDFs fitted both the E866 deuteron-to-proton ratios as well as the absolute pp cross sections. As explained in the text following the second sentence indicated by the referee above, CT18 adopted a PDF parametrization that favored dbar/ubar > 1 at high x, as is expected based on nonperturbative QCD models. As such, before fitting, CT18 is already consistent with the reported SeaQuest ratio data. Of course, even from the SeaQuest main paper itself it is clear that the very highest 1-2 x_t bins measured by E866 differ from the new SeaQuest data. As such, when the two data sets are fitted simultaneously, there is a clear tension between SeaQuest and E866 at the highest-x values for dbar/ubar. Still, this tension originates with a select number of high-x_t points such that, overall, the fit decently describes both experiments. The question of how best (or whether) to fit both experiments is a question that we begin to investigate in these proceedings; ultimately settling this issue requires further analysis along the lines of a forthcoming CT paper. For now, we have slightly expanded the text (with additions indicated by [...]) around the second sentence indicated by the referee above (i.e., on Page 3) to now read:

"We find the SeaQuest data to be in [overall] agreement with theory predictions based on CT18 NNLO {\it before fitting}, with chi2 / N_{pt} = 0.82. This partly reflects the parametrization choices made in CT18~\cite{Hou:2019efy} for the high-$x$ behavior of $\bar{d}, \bar{u}$, which are selected to preserve $\bar{d}/\bar{u} \ge 1$ at high $x$ on the QCD modeling logic discussed above. [Unlike the highest $x_t$ E866 ratio data, t]he [newer] SeaQuest ratios thus prefer the $\bar{d}/\bar{u} \ge 1$ high-$x$ behavior favored by nonperturbative QCD-motivated models discussed above."

8- Figures 3-4. Can the authors please use the same symbols/colours for the same PDF sets displayed in the two figures?

--> The symbols and colors have now been uniformized between Figs. 3/4.

9- Page 7, first line of Sect. 4. Can the authors clarify the difference between x , x t and x B ? The three notations are used across the manuscript.

--> We have added a footnote briefly discussing these distinctions at the start of Sec. 4. Together with the definition of x_t following Eq. (1), this point has been clarified.

10- I believe that I've spotted two typos. 1st line of page 4: simulataneously -> simultaneously; 26th line of page 8: deceased -> decreased.

--> Thank you for pointing these out; they have been fixed. \end{document}

---

## Editorial Decision

resubmitted